# SOAT: A Scene- and Object-Aware Transformer for Vision-and-Language Navigation

**Abhinav Moudgil**[1]*, **Arjun Majumdar**[1], **Harsh Agrawal**[1], **Stefan Lee**[2], **Dhruv Batra**[1]
[1] Georgia Institute of Technology, [2] Oregon State University

## Abstract

Natural language instructions for visual navigation often use *scene descriptions* (e.g., *'bedroom'*) and *object references* (e.g., *'green chairs'*) to provide a breadcrumb trail to a goal location. This work presents a transformer-based vision-and-language navigation (VLN) agent that uses two different visual encoders – a scene classification network and an object detector – which produce features that match these two distinct types of visual cues. In our method, scene features contribute high-level contextual information that supports object-level processing. With this design, our model is able to use vision-and-language pretraining (i.e., learning the alignment between images and text from large-scale web data) to substantially improve performance on the Room-to-Room (R2R) [1] and Room-Across-Room (RxR) [2] benchmarks. Specifically, our approach leads to improvements of 1.8% absolute in SPL on R2R and 3.7% absolute in SR on RxR. Our analysis reveals even larger gains for navigation instructions that contain six or more object references, which further suggests that our approach is better able to use object features and align them to references in the instructions.

## 1   Introduction

The vision-and-language navigation (VLN) task [1] requires an agent to follow a path through an environment that is specified by natural language navigation instructions. A central component of this task is associating (or grounding) the instruction to visual landmarks in the environment. Figure 1 provides an illustrative example from the Room-to-Room (R2R) dataset [1]: *'Exit the bedroom and turn left. Continue down the hall and into the room straight ahead and stop before the desk with two green chairs.'* The visual landmarks in this instruction include scene descriptions (e.g., *'bedroom'* and *'hall'*) and specific object references (e.g., *'desk'* and *'two green chairs'*). To be successful, a VLN agent should be able to (a) recognize and (b) ground both types of visual cues.

Learning the appropriate grounding between referring expressions in an instruction and the corresponding visual regions is difficult in VLN due to the limited visual diversity seen in training. For example, the Room-to-Room (R2R) [1] and Room-Across-Room (RxR) [2] datasets only use 61 unique training environments, so models simply cannot learn about the long-tail of visual cues that appear in new testing (or validation) scenes. To address this challenge, recent work has shown the promise of transferring visual grounding with multimodal representation models that are pretrained on a large amount of image-text web data before finetuning on the embodied VLN task [3, 4]. In this work, we build on this general approach.

For visual recognition, most VLN methods [1, 4–14] first encode observations with a convolutional network that was trained to solve an image-level classification task – either using ImageNet [15] or the Places [16] scene recognition dataset. While ImageNet features may identify objects mentioned in the instructions and Places features might match the scene descriptions, neither solution was

---

*Correspondence to `abhinavmoudgil95@gmail.com`

35th Conference on Neural Information Processing Systems (NeurIPS 2021).

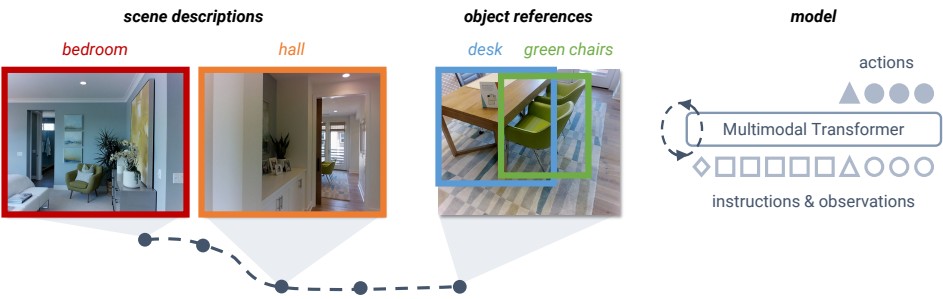

*scene descriptions*     *object references*     *model*

Navigation Instructions: *"Exit the bedroom and turn left. Continue down the hall and into the room straight ahead and stop before the desk with two green chairs."*

Figure 1: Illustrative VLN Instruction. Visual landmarks in the instruction include scene descriptions (e.g., *'bedroom'* and *'hall'*) and object references (e.g., *'desk'* and *'green chairs'*). Unlike most previous methods, our model uses both scene (△) and object features (◯) to match the visual references in the instructions. We employ a novel attention mechanism and view aggregation strategy to select the most relevant features for action prediction.

explicitly trained to recognize both types of visual cues, which is a limitation that we address in this paper. Furthermore, pretrained multimodal representation models have typically been trained with features from object detection networks rather than the image-level representations commonly used in VLN. Despite this, prior work leveraging these models [4] has continued to use the standard set of image-level features – leading to a significant domain shift between pretraining and VLN finetuning.

In this work, we address these issues 1) by using multiple visual encoders to explicitly encode the inductive bias that the world is composed of objects and scenes and 2) by using object-level features from a detection network that match the features used to pretrained multimodal representation models. The starting point for our work is a VLN⟳BERT [4] agent that processes scene-level features from a Places [16] CNN with a transformer-based [17] multimodal representation model that is modified with a recurrence mechanism designed for the VLN task. Our work extends this approach by including object features as an additional input to the multimodal processing. However, we find that simply adding object features to VLN⟳BERT does not improve (but rather slightly reduces) performance. Thus, we propose architectural changes that allow the model to take better advantage of these two distinct types of visual information. Specifically, we change the attention pattern within the the transformer to effectively freeze the scene representations and focus the processing on the object-level inputs. The result is a new VLN agent that produces contextualized object representations by using scene features as high-level contextual cues.

We experiment with our proposed approach on the Room-to-Room (R2R) [1] and Room-Across-Room (RxR) [2] datasets. Empirically, we find that our model substantially improves VLN performance over our VLN⟳BERT baseline on R2R and outperforms state-of-the-art methods on English language instructions in RxR. Specifically, our proposed approach improves success weighted by path length (SPL) on the unseen validation split in R2R by 1.8 absolute percentage points. On RxR – a more challenging dataset due to indirect paths and greater variations in path length – we see even larger improvements. Success rate (SR) improves by 3.7 absolute percentage points, alongside a gain of 2.4 absolute percentage points on the normalized dynamic time warping (NDTW) metric. Through ablation experiments we find that (consistent with the observations in [3]) vision-and-language pretraining is vital to our approach, which suggests that strong visual grounding is key for using object-level features in VLN. Additionally, on RxR instructions that include six or more object references (i.e., object-heavy instructions), our method has even larger improvements over VLN⟳BERT of 7.9 absolute percentage points in SR.

To summarize, we make the following contributions:

- We propose a scene- and object-aware transformer (SOAT) model for vision-and-language navigation that uses both scene-level and object-level features – explicitly encoding the inductive bias that the world is composed of objects and scenes. Our model uses a novel attention masking technique and view aggregation strategy, which both improve performance.

- We demonstrate that our approach outperforms a strong baseline approach by 1.8 absolute percentage points in SPL on R2R and by 3.7 absolute percentage points in SR on RxR.
- We show that our method has significantly stronger performance on instructions that mention six or more objects (7.9 absolute percentage points of improvement in SR), which further suggests that our model is better able to recognize and ground object references.

## 2 Related Work

**Vision-and-Language Navigation.** The Room-to-Room (R2R) [1] and Room-Across-Room (RxR) [2] datasets both situate the VLN task within Matterport3D [18] indoor environments. Since the release of R2R there has been steady improvement in VLN task performance [4–14]. Some of the key innovations include using instruction-generation via *'speaker'* models for data augmentation [5, 10], combining imitation and reinforcement learning [6], using auxiliary losses [7, 13], and different pretraining strategies [4, 12, 14]. All of these methods have one thing in common – they process visual observations with a single convolutional network pretrained to solve an classification task (using either the ImageNet [15] or Places [16] datasets). In contrast, this work explores using a combination of features from visual encoders pretrained for scene classification and object detection.

**Object Detectors in VLN.** Intuitively, object detections should naturally match the object cues (e.g., *'green chairs'*) mentioned in VLN instructions. Indeed, several recent studies [3, 19–21] have demonstrated the utility of using object detectors for VLN. In [19, 20] object classification labels from an object detector are encoded using a GLoVe [22] embedding. Similarly, [21] convert detections into a feature vector using the classification label, object area, and detector confidence. Unlike these methods, our model directly uses object features produced by a detector, which provide a richer, high-dimensional representation of each region. Additionally, our approach takes advantage of vision-and-language pretraining, which eases the burden of learning the grounding between natural language and object representations from scratch using only VLN data, as is done in these prior methods. In [3], object features are used in a model that solves a path selection task in VLN, which requires pre-exploring an environment before executing the navigation task. By comparison, this work focuses on navigating without pre-exploration.

## 3 Preliminaries: VLN, Visual Encoders, Multimodal Transformers

This section reviews the vision-and-language navigation (VLN) task and describes how multimodal transformers are used in the recently proposed VLN↻BERT [4] model. Our approach builds on VLN↻BERT and is described in Section 4.

### 3.1 Vision-and-Language Navigation

In VLN, agents are placed in a photo-realistic 3D environment and must navigate to a goal location that is specified through natural language navigational instructions $I$ (illustrated in Figure 1). At each timestep $t$, the agent receives a set of panoramic observations $O_t = \{o_{t,i}\}_{i=1}^{36}$ composed of RGB images from 36 viewing angles (12 headings $\times$ 3 elevations). We follow the VLN with known navigation graph setting [1, 2] – agents have access to a graph that specifies a set of navigable locations from each viewpoint in the environment. We agree with the limitations of this setting discussed in [23]. However, we report results for the nav-graph setting to be consistent with the long line of prior work in this area. We plan on generalizing to continuous environments in the future. Using the nav-graph, agents select an action from the set $A_t = \{a_{t,i}\}_{i=0}^{N_t}$ consisting of $N_t$ navigable locations and the `stop` action. The agent is successful if it calls `stop` within 3m of the goal location.

### 3.2 Multimodal Transformers

Here we provide a brief overview of multimodal transformers (e.g., OSCAR [24]), which provide a basis for the VLN↻BERT architecture. Multimodal transformers are an extension of transformer-based [17] language models such as BERT [25] that process image-text pairs. As in BERT, the text input is tokenized, encoded with a learned embedding, and then combined with positional information (i.e., word order). Commonly, the visual input (i.e., the image) is first preprocessed by an object

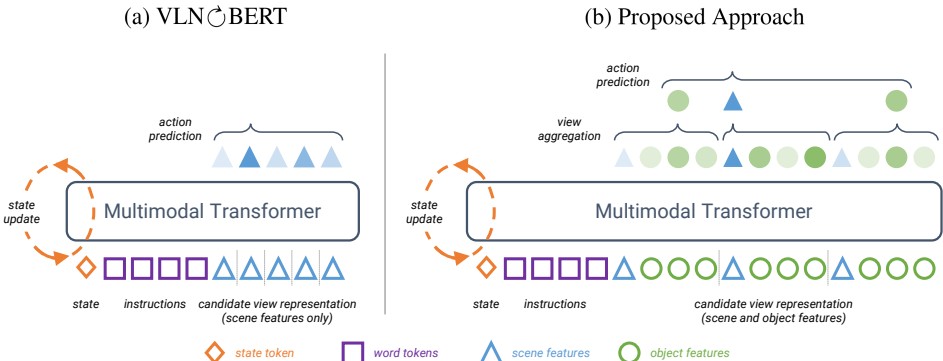

Figure 2: Model architectures. (a) VLN↻BERT uses a multimodal transformer to encode state history, instruction tokens and scene features from candidate views. The candidate view corresponding to the scene features with the highest attention score is chosen as the next action. (b) Our approach *includes object features* as additional input. In view aggregation, the visual feature (either object or scene) with the maximum attention score is selected to represent each candidate view. For action prediction, the model chooses the candidate view with the highest attention score as the next action.

detector (e.g., a Faster R-CNN [26] model trained on Visual Genome [27]) to produce a set of region features that are combined with spatial information (e.g., the offset to the bounding box in the image). To summarize, the multimodal input consisting of $L$ word tokens $\{w_1, \ldots, w_L\}$ and $M$ image regions $\{r_1, \ldots, r_M\}$ can be written as

$$[\texttt{CLS}], w_1, \ldots, w_L, [\texttt{SEP}], r_1, \ldots, r_M$$

where $[\texttt{CLS}]$ is a special token used as a global representation of the input and the $[\texttt{SEP}]$ token separates modalities. This set of multimodal inputs are fed to a series of transformer encoder layers [17] to produce contextualized representations for each input using attention-based processing.

## 3.3 Multimodal Transformers for VLN

The input interface for multimodal transformers can be easily adapted to process the language instructions and panoramic image from each timestep in VLN. However, VLN requires sequentially following navigation instructions (e.g., *'Exit the bedroom...'* then *'Continue down the hall...'* then *'stop before the desk with two green chairs.'*). Accordingly, maintaining a history of the agent's state is helpful for understanding which sub-instruction to follow at each timestep. Traditional VLN agents use recurrent neural networks (e.g., LSTM [28]) to model state history. In contrast, VLN↻BERT [4] introduces a generic recurrence mechanism that can, in principle, be added to any multimodal transformer model to refashion it for the VLN task. In this work we extend the OSCAR [24] instantiation of VLN↻BERT.

As shown in Figure 2a, VLN↻BERT [4] uses a multimodal transformer to process word tokens from the navigation instructions and scene features for a set of "candidate" views from a panoramic observation. There is one candidate view for each navigable action in $\boldsymbol{A}_t$, corresponding with the views that best align with each navigable direction (there are ∼4 such views on average). Accordingly, the output representation for each view is used to compute the probability of moving in that direction. The recurrence mechanism in VLN↻BERT is operationalized using a state token $\boldsymbol{s}_t$, which is fed as an input to the multimodal transformer and is updated at each timestep.

**Initialization.** The state token $\boldsymbol{s}_t$ is initialized by passing word tokens from the navigation instruction $\boldsymbol{I}$ along with special $[\texttt{CLS}]$ and $[\texttt{SEP}]$ tokens through the multimodal transformer. The output representation for the $[\texttt{CLS}]$ token is used to set $\boldsymbol{s}_0$ and the outputs for the word tokens (denoted as $\psi(\boldsymbol{I})$) are used as the language representation during navigation.

**Visual Inputs.** For each RGB image in the $N_t$ candidate views, VLN↻BERT runs a convolutional network trained on Places [16] to extract high-level scene features. To facilitate the $\texttt{stop}$ action an all-zeros feature vector is added to the set of visual inputs. We denoted this set of visual inputs as $\boldsymbol{V}_t$.

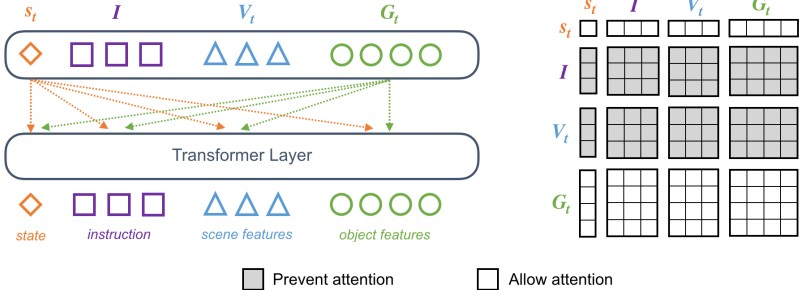

Figure 3: Selective attention: during navigation the instruction and scene features are kept frozen – i.e., they are not updated in the transformer layers and just serve as keys and values. The state token and object features are updated through self-attention by attending to *all* input tokens.

**Navigation.** At each timestep, the input to the multimodal transformer is composed of the state token $s_t$, the encoded instruction $\psi(\boldsymbol{I})$, and the visual inputs $\boldsymbol{V}_t$. To reduce computation, the instruction representation is not updated during navigation; the encoded word tokens simply serve as keys and values when processed by the multimodal transformer.

Actions are predicted using "state-conditioned" attention scores for the scene features. These scores are produced at the final layer of the multimodal transformer. Concretely, let the final layer outputs for the state token $s_t$ and scene features $\boldsymbol{V}_t$ be denoted as $\psi(\boldsymbol{s}_t) \in \mathbb{R}^d$ and $\psi(\boldsymbol{V}_t) \in \mathbb{R}^{(N_t+1) \times d}$ where $N_t$ is the number of navigable locations and $d$ is the model's hidden dimension size. State-conditioned attention scores are calculate using a scaled dot-product as

$$\alpha(\boldsymbol{V_t}) = \frac{\psi(\boldsymbol{s}_t)\psi(\boldsymbol{V}_t)^T}{\sqrt{d}} \tag{1}$$

These scores $\alpha(\boldsymbol{V_t})$ are normalized using the softmax function $\tilde{a}(\boldsymbol{V_t}) = \texttt{softmax}(\alpha(\boldsymbol{V_t}))$ and used as the probability of moving to one of the $N_t$ navigable locations or stopping. Finally, the state history is maintained by constructing the next state token $\boldsymbol{s}_{t+1}$ with the output representation for the state token $\psi(\boldsymbol{s}_t)$ using a "refinement" procedure detailed in the Appendix.

## 4    Approach

Motivated by the observation that VLN instructions often use scene descriptions (e.g., *'bedroom'*) and object references (e.g., *'green chairs'*) to provide visual cues, our goal is to design a model that effectively uses scene and object features for VLN. One straightforward idea is to add an object feature input to VLN↻BERT [4]. However, we find that this simple solution does not improve VLN performance (as we show in Table 3 row 2), which is perhaps why object features were not used for navigation in [4].[2] Thus, we redesigned the VLN↻BERT architecture to create a scene- and object-aware transformer (SOAT) model that benefits from access to both scene and object features.

First, we modify the processing in the model to focus on object features by designing the attention mask shown in Figure 3 and described in Section 4.2. The mask only allows the state token and object features to be updated by the multimodal transformer. Second, as illustrated in Figure 2b, the output from our model includes scene and object representations for each candidate view. Thus, we design a simple approach (discussed in Section 4.3) to aggregate this information into a single representation for each view, which is used for action prediction.

### 4.1    Visual Inputs

To generate scene and object features we use two different visual encoders to process RGB images from the panoramic observations. Matching VLN↻BERT, we use a CNN trained on the Places [16] dataset to extract scene features for each candidate view, and use an all-zeros feature vector for the `stop` action to produce the set of scene features $\boldsymbol{V}_t$. For object features, we use the same object

---

[2]Object features were used in [4] to select objects for a referring expressions task [29] but not for navigation.

detector that was used to pretrain the multimodal transformer (i.e., OSCAR [24]) that we use to initialize our model – avoiding any additional domain gap between pretraining and fine-tuning. Specifically, we use a Faster R-CNN [26] detector trained on Visual Genome [27] using the training procedure from [30]. We denote the set of object features for detections from all of the candidate views as $G_t$. Note that for a given image the detector may or may not detect any objects. As a result, in our model, each candidate view is represented with one scene feature and zero or more object feature vectors. As illustrated in Figure 2b, the full set of inputs to our model include the state token $s_t$, encoded instructions $\psi(I)$, and the visual inputs consisting of scene $V_t$ and object $G_t$ features.

## 4.2 Selective Object Attention

Simply adding object features as an additional input to the model does not improve performance. To overcome this challenge, we adjust the processing to focus on the object feature inputs. This is accomplished using the attention mask show in Figure 3 (right). Like VLN$\circlearrowright$BERT, with this attention mask, the encoded navigation instructions $\psi(I)$ are not updated in the multimodal transformer. However, in our approach the scene features $V_t$ are also kept frozen. As a result, these inputs only serve as keys and values during the attention based processing. The state token $s_t$ and object features $G_t$ operate as queries, keys and values, which is standard in transformer-based models.

Intuitively, this attention mask allows scene features to provide contextual information to support refining the object representations. Note that the (unaltered) scene features are still used for action prediction (described in Section 4.3), which provides flexibility in how the model uses both types of visual representations. We hypothesize that this design leads to more effective transfer learning (demonstrated in Section 5), because the object-centric processing more closely matches the pretraining setup in which the model only operates on language and object features.

## 4.3 View Aggregation

At each timestep of navigation, the multimodal transformer uses selective object attention (described above) to produce contextualized representations of the state $\psi(s_t)$ and object features $\psi(G_t)$. As illustrated in Figure 2b, the object representations $\psi(G_t)$ alongside the unaltered scene features $V_t$ are used for view aggregation and action prediction. Specifically, state-conditioned attention scores are calculated for the scene and object features using a scaled dot-product (as in Equation (1)) as

$$\alpha(V_t) = \frac{\psi(s_t)V_t^T}{\sqrt{d}}, \qquad \alpha(G_t) = \frac{\psi(s_t)\psi(G_t)^T}{\sqrt{d}} \qquad (2)$$

where $d$ is the model's hidden dimension size. As a result, each candidate view is represented with one scene feature attention score and zero or more object feature attention scores.

For view aggregation, we select the maximum attention score for all of the scene and object features in a given view, and use the corresponding visual input to represent that view. Intuitively, this approach allows the model to either select a relevant object (which may be mentioned in the instructions such as *'green chairs'*) or the full scene (which might match scene descriptions such as *'hallway'*) to represent each navigable viewpoint (i.e., candidate view). Finally, for action prediction, we take the softmax over the selected attention scores and use these normalized scores to represent the probability of moving to each navigable location or stopping.

# 5 Experiments

**Datasets.** We evaluate our method on the Room-to-Room (R2R) [1] and Room-Across-Room (RxR) [2] datasets. R2R is built using Matterport3D (MP3D) [18] indoor scenes and contains 21,567 path-instruction pairs, which are divided into four splits: training (14,025), val-seen (1,020), val-unseen (2,349) and test-unseen (4,173). Val-seen uses environments from the training split but the path-instruction pairs are novel. Val-unseen and test-unseen use new environments and new path-instruction pairs to evaluate generalization performance. We augment the R2R training data with 1M instructions generated by the speaker model from [14].

RxR [2] is a recently introduced multi-lingual VLN dataset that also uses MP3D scenes. It contains 126K path-instruction pairs in 3 languages (Hindi, English and Telugu). English instructions are collected from two regions: India (en-IN) and US (en-US). Since our model is pretrained on English

Table 1: R2R and RxR results on val-unseen. Rows 1 and 2 indicate performance of the random and human baselines. Rows 3-7 are results from prior state-of-the-art methods. Rows 8 and 9 provide results of VLN↻BERT [4] and our method in the same setting. † indicates reproduced results.

| | Methods | Pretraining | | R2R | | | | RxR | | | |
|---|---|---|---|---|---|---|---|---|---|---|---|
| | | Web | VLN | TL | NE ↓ | SR ↑ | SPL ↑ | NE ↓ | SR ↑ | SDTW ↑ | NDTW ↑ |
| 1 | Random | | | 9.77 | 9.23 | 16 | - | 9.5 | 5.1 | 3.8 | 27.6 |
| 2 | Human | | | - | - | - | - | 1.32 | 90.4 | 74.3 | 77.7 |
| 3 | RxR baseline[2] | | | - | - | 37 | 32 | 10.1 | 25.6 | 20.3 | 41.3 |
| 4 | EnvDrop [10] | | | 10.70 | 5.22 | 52 | 48 | - | - | - | - |
| 5 | PREVALENT [14] | | ✓ | 10.19 | 4.71 | 58 | 53 | - | - | - | - |
| 6 | VLN↻BERT [4] (init. OSCAR) | ✓ | | 11.86 | 4.29 | 59 | 53 | - | - | - | - |
| 7 | VLN↻BERT [4] (init. PREVALENT) | | ✓ | 12.01 | **3.93** | **63** | **57** | - | - | - | - |
| 8 | VLN↻BERT [4] (init. OSCAR) † | ✓ | | 12.16 | 4.40 | 58 | 51 | 7.31 | 40.5 | 33.0 | 52.4 |
| 9 | Ours | ✓ | | 12.15 | **4.28** | 59 | 53 | 6.72 | **44.2** | **36.4** | **54.8** |

language data, we focus on the English language subset of RxR (both `en-IN` and `en-US`), which includes 26,464 path-instruction pairs for training and 4,551 pairs in the val-unseen split.

**Evaluation.** We follow the standard evaluation protocols for R2R and RxR as shown in Table 1. When applicable, we use an ↑ to indicate higher is better and a ↓ to indicate lower is better. On R2R, we report: Trajectory Length (TL), Navigation Error (NE ↓) - average distance between the target and the agent's final position, Success Rate (SR ↑) - percentage of trajectories in which agent stopped within 3 meters of the target, and Success weighted by normalized inverse of Path Length (SPL ↑). On RxR dataset, in addition to the NE and SR metrics, we also report the Normalized Dynamic Time Warping (NDTW ↑) and Success weighted by normalized Dynamic Time Warping (SDTW ↑) metrics, which explicitly measure path adherence. Additional details are included in the Appendix.

**Baselines.** In Table 1, we compare our approach (Row 9) with recent state-of-the-art methods. EnvDrop [10] (Row 4) trains an encoder-decoder model [5] on augmented data (in addition to R2R training data) generated with back-translation using "dropped out" scenes in order to generalize well to unseen environments. PREVALENT [14] (Row 5) builds on EnvDrop by pretraining a multimodal transformer model on augmented data. After pretraining, PREVALENT feeds contextualized word embeddings from this pretrained transformer into the EnvDrop [10] encoder-decoder to fine-tune on R2R dataset. VLN↻BERT (Row 6-8) extends PREVALENT by directly fine-tuning the pretrained multimodal transformer for navigation. VLN↻BERT uses two initializations – OSCAR (Row 6) and PREVALENT (Row 7). For fair comparison, we also report results reproduced with the released implementation[3] of VLN↻BERT by training from the OSCAR initialization (Row 8). Importantly, all these methods use *either* scene or object features as visual inputs for navigation.

**Implementation Details.** We implemented our model in PyTorch [31] and trained on a single Nvidia TitanX GPU. Consistent with VLN↻BERT [4], we initialize our model with a pretrained OSCAR model [24]. For both R2R and RxR, we fine-tune with behaviour cloning and reinforcement learning (RL) objectives adapted from prior work [4, 10, 14]. As in [4], 50% of each batch consists of behaviour cloning rollouts and 50% from RL (policy gradient). We train with a constant learning rate of 1e-5 using the AdamW optimizer with a batch size of 16 for 300k iterations. Like VLN↻BERT, we extract scene features with a ResNet-152 model [32] pretrained on the Places dataset [16]. For object features, we extract detections from a pretrained bottom-up attention model [30], and adopt the filtering procedure from [3] to discard redundant detections. In all of our experiments we use the same hyperparameters for the VLN↻BERT baseline and our approach.

## 5.1 Main Results

**Performance on R2R and RxR.** Rows 8 and 9 of Table 1 provide a comparison with the VLN↻BERT baseline on R2R (left) and RxR (right). RxR is a more challenging dataset because there is more variation in path lengths and agents often need to follow indirect paths to a goal. On RxR, our approach (row 9), which uses both scene and object features, beats the scene features only VLN↻BERT baseline (row 8) by **3.7%** on SR, **3.4%** on SDTW and **2.4%** on NDTW. On R2R, our approach (row 9) provides a **1.8%** gain in SPL over the VLN↻BERT baseline (row 8). The larger

---

[3] `https://github.com/YicongHong/Recurrent-VLN-BERT`

Table 2: R2R test results. † indicates reproduced results.

| | Methods | Pretraining | | R2R Test Unseen | | | |
|---|---|---|---|---|---|---|---|
| | | Web | VLN | TL | NE ↓ | SR ↑ | SPL ↑ |
| 1 | Random | | | 9.89 | 9.79 | 13 | 12 |
| 2 | Human | | | 11.85 | 1.61 | 86 | 76 |
| 3 | EnvDrop [10] | | | 11.66 | 5.23 | 51 | 47 |
| 4 | PREVALENT [14] | | ✓ | 10.51 | 5.30 | 54 | 51 |
| 5 | VLN○BERT [4] (init. OSCAR) | ✓ | | 12.34 | 4.59 | 57 | 53 |
| 6 | VLN○BERT [4] (init. PREVALENT) | | ✓ | 12.35 | **4.09** | **63** | **57** |
| 7 | VLN○BERT [4] (init. OSCAR) † | ✓ | | 12.78 | 4.55 | **58** | 52 |
| 8 | Ours | ✓ | | 12.26 | **4.49** | **58** | **53** |

gains on RxR may result from a greater number of object references in RxR instructions (which is suggested by the linguistic analysis in [2]), which are required to describe the more complex paths. This hypothesis is consistent with our analysis in section 5.2 that shows larger gains from our method on instructions that contain 6 or more object references. These significant improvements highlight the benefit of using object features for challenging datasets like RxR.

Table 1 rows 3-7 show the results for state-of-the-art methods on R2R and RxR. On RxR, our method outperforms the previous state-of-the-art established by the RxR baseline (row 3) by **13.5%** absolute on NDTW and **18.6%** absolute on SR. On R2R, our approach is competitive with the previous state-of-the-art across all metrics. Table 2 reports results on the R2R test-unseen split, which again shows that our approach is competitive with prior work. In both tables VLN○BERT with a PREVALENT initialization outperforms our approach. However, we note that PREVALENT [14] uses an alternative pretraining setup with scene features and augmented VLN data, which is orthogonal to our goals of using object features and transferring visual grounding for VLN. Combining these two different pretraining strategies is an interesting direction for future work.

## 5.2 Ablations and Analysis

Table 3: Ablation Study on R2R. † indicates reproduced results.

| | Models | Object Features | View Aggregation | Selective Attention | Val Unseen | |
|---|---|---|---|---|---|---|
| | | | | | SR ↑ | SPL ↑ |
| 1 | Baseline [4]† | | | | 57.90 | 51.43 |
| 2 | | ✓ | | | 57.26 | 50.96 |
| 3 | | ✓ | ✓ | | 57.85 | 51.73 |
| 4 | | ✓ | | ✓ | 57.73 | 52.46 |
| 5 | | | | ✓ | 57.60 | 52.06 |
| 6 | Ours | ✓ | ✓ | ✓ | **58.71** | **53.24** |

**Do object features help without the proposed architectural changes?** Table 3 presents the results of an ablation experiment on R2R that demonstrates the importance of each design choice in our model. In row 2 object features are used as an additional input to VLN○BERT with no other architectural changes (e.g., selective attention or view aggregation). In this ablation, the visual inputs (scene and object features) all get refined with self-attention and action probabilities are calculated in a similar fashion to VLN○BERT [4]. We see that merely using object features as an additional input leads to slightly reduced performance on val-unseen (rows 1 vs. 2). Rows 3 and 4 demonstrate that using object features with either view aggregation (Section 4.3) or selective attention (Section 4.2) reverses this trend. Furthermore, row 6 demonstrates that the largest gains are seen when all three modifications to the VLN○BERT baseline are used together – resulting in a **1.8%** gain in SPL. Additionally, row 5 shows that using selective attention without object features does not out perform our full approach. More details are provided in the appendix. To summarize, using object features only leads to improved performance with the architectural changes proposed in this work.

Table 4: Results on R2R without vision-and-language pretraining.

| | Models | R2R Val Unseen | | | |
|---|---|---|---|---|---|
| | | TL | NE ↓ | SR ↑ | SPL↑ |
| 1 | VLN↻BERT [4] | 10.31 | 5.21 | 49.21 | 45.53 |
| 2 | Ours | 12.00 | 4.99 | 50.36 | 44.69 |

**Do object features help without vision-and-language pretraining?**   Table 4 reports results of the VLN↻BERT baseline and our approach without vision-and-language pretraining on the R2R val-unseen split. As evident from Table 4, both the approaches perform similarly across all metrics. This demonstrates that transferring visual grounding is crucial for our approach to work; our approach improves performance compared to the VLN↻BERT baseline (Table 1 Row 7 vs Row 8) predominantly because it more effectively utilizes the large-scale vision-and-language pretraining.

Table 5: Results on a subset of RxR instructions divided by the number of object references.

| | Models | Object Heavy Instructions | | | | Not Object Heavy Instructions | | | |
|---|---|---|---|---|---|---|---|---|---|
| | | NE ↓ | SR ↑ | SDTW ↑ | NDTW ↑ | NE ↓ | SR ↑ | SDTW ↑ | NDTW ↑ |
| 1 | VLN↻BERT [4] | 8.46 | 33.89 | 26.36 | 46.95 | 4.07 | 58.21 | 50.33 | 67.75 |
| 2 | Ours | **7.27** | **41.80** | **32.92** | **52.46** | **3.77** | **61.64** | **53.87** | **69.37** |

**How does our method perform on instructions that heavily mention objects?**   The RxR dataset contains long instructions that frequently mention objects and scenes. Specifically, RxR contains 6 entity references per instruction on average, which is nearly double the amount of entities referenced in R2R (3.7) [2]. Here, we evaluate our model on a subset of RxR instructions that heavily contain object references. To find this subset, we first randomly pick 500 instructions from val-unseen split in RxR. Then, we extract noun entities from the instructions using a standard NLP parser [33]. Next, we manually inspect these nouns and remove direction, scene, or other non-object references. This results in a list that only contains object references. Finally, we split the randomly selected 500 instructions into two sets containing 6 or more object object references (i.e., object heavy instructions) or less than 6 object references (i.e., not object heavy instructions). The threshold of 6 was selected to match the average number of entity references in RxR instructions [2]. With this process 354 of the 500 instructions were selected for object heavy set.

Results of our method and VLN↻BERT baseline on these subsets are reported in Table 5. On the object heavy subset, our approach gives *even stronger performance improvements* over VLN↻BERT baseline than seen on the full RxR val-unseen split. Specifically, we obtain an absolute improvement of **7.91%** for Success Rate (SR), **6.56%** for SDTW score and **5.51%** for NDTW. These gains are approximately twice the improvement on val-unseen split and larger that the ∼**3%** improvements on the complementary subset containing 146 instructions with less that 6 object references (Table 5 right). These results suggest that our approach is able to exploit object features and appropriately align them with object references to follow visually grounded navigation instructions.

## 6   Conclusion

In this work, we propose a novel scene- and object-aware transformer (SOAT) model, which uses scene and object features for vision-and-language navigation. We introduce a selective attention pattern in the transformer for processing these two distinct types of visual inputs such that scene tokens only provide contextual information for object-centric processing. As a result, our approach improves overall performance over a strong baseline on the R2R and RxR benchmarks. Our analysis shows that vision-and-language pretraining is crucial for exploiting the newly added object features. We obtain even larger performance improvement on instructions that heavily mention objects, which suggests that our approach is able to effectively ground object references in the instructions to the visual object features.

# 7 Acknowledgements

The Georgia Tech effort was supported in part by NSF, ONR YIPs, ARO PECASE, Amazon. The views and conclusions contained herein are those of the authors and should not be interpreted as necessarily representing the official policies or endorsements, either expressed or implied, of the U.S. Government, or any sponsor.

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
