# SOAT: A Scene- and Object-Aware Transformer for Vision-and-Language Navigation

**Abhinav Moudgil**[1]*, **Arjun Majumdar**[1], **Harsh Agrawal**[1], **Stefan Lee**[2], **Dhruv Batra**[1]

[1] Georgia Institute of Technology, [2] Oregon State University

## A  Appendix

### A.1  Limitations

We propose an approach which exploits object features in addition to scene features for vision-and-language navigation (VLN). Our approach is able to utilize object features for better visiolinguistic alignment (see Section 5) despite the domain gap between the images used to train the object detector and VLN data. Specifically, object features are obtained using a Faster R-CNN detector [1] trained on photos from web (Visual Genome [2]), in which objects are typically well framed by the photographer. On the other hand, the VLN datasets used in our experiments contain panoramic images from indoor house scans that capture objects at viewing angles determined by the navigation path. The gap between these two types of data could be eliminated by either fine-tuning or training detector directly on indoor scenes. This domain gap is also present during pretraining. Our approach uses an OSCAR model [3] that was pretrained on image and text pairs from the internet. An additional pretraining stage on VLN data (e.g., by adapting the pretraining techniques proposed in [4]) may further improve performance, and is an interesting direction for future work. Finally, we test our approach on English instructions from RxR. However, future work might explore how to extend our approach for the multi-lingual setting.

### A.2  Broader Impact

We propose a new model with better vision-and-language navigation performance in indoor environments. This work is a step towards building AI agents which can follow natural language instructions from humans and act accordingly. These AI agents could be deployed in home and commercial environments to provide assistance. However, since these agents are trained using the path-instruction pairs from VLN datasets, they encode all the visual and language biases that are present in the data (e.g., types of houses, objects, language usage, etc). Hence, significant attention should be paid before deployment of these agents in the real-world.

### A.3  Performance for Multiple Random Seeds

We report results with 3 random seeds of our approach and VLN↻BERT baseline in Table 1 on R2R dataset [5]. We report the mean and standard error for each metric. Overall, our approach improves SPL by $\sim 1\%$ which is consistent with the reported results in the main draft.

### A.4  Additional Analysis of Selective Object Attention

In Table 2 we report results (on R2R) for alternatives to the selective object attention mask proposed in this work. Specifically, we vary which inputs only serve as keys and values and which serve as queries, keys, and values in the multimodal transformer. Results for our model with selective object attention are presented in row 5. The alternative of performing selective *scene* attention

---

*Correspondence to `abhinavmoudgil95@gmail.com`

35th Conference on Neural Information Processing Systems (NeurIPS 2021).

Table 1: Results with 3 random seeds.

| | Models | R2R Val Unseen | | | |
|---|---|---|---|---|---|
| | | TL | NE ↓ | SR ↑ | SPL ↑ |
| 1 | VLN↻BERT [6] | $11.76 \pm 0.29$ | $4.44 \pm 0.04$ | $57.37 \pm 0.69$ | $52.30 \pm 0.33$ |
| 2 | Ours | $11.25 \pm 0.08$ | $4.32 \pm 0.06$ | $58.01 \pm 0.39$ | $53.37 \pm 0.33$ |

(row 2) underperforms our approach. Similarly, as discussed in Section 5, the performance of our model without selective object attention (row 3) is comparable to the VLN↻BERT baseline. Row 4 reports results of only using object features for the visual inputs, which leads to substantially lower performance than our approach of using scene and object features. Since our approach (row 5) is closely aligned with pretraining setup, it is able to utilizes both object- and scene-level visual cues to outperform all of these alternatives.

Table 2: R2R attention ablations.

| | Models | Input Tokens | Query Tokens | Val Seen | | | | Val Unseen | | | |
|---|---|---|---|---|---|---|---|---|---|---|---|
| | | | | TL | NE ↓ | SR ↑ | SPL ↑ | TL | NE ↓ | SR ↑ | SPL ↑ |
| 1 | Baseline [6] | $\langle s, X, V \rangle$ | $\langle s, V \rangle$ | 11.02 | 3.11 | 69.93 | 65.65 | 12.16 | 4.40 | 57.90 | 51.43 |
| 2 | Scene attention | $\langle s, X, V, O \rangle$ | $\langle s, V \rangle$ | 11.08 | 3.35 | 68.95 | 64.48 | 12.68 | 4.42 | 56.83 | 50.99 |
| 3 | All attention | $\langle s, X, V, O \rangle$ | $\langle s, V, O \rangle$ | 11.12 | 3.24 | 70.03 | 66.59 | 12.05 | 4.34 | 57.85 | 51.73 |
| 4 | Object attention | $\langle s, X, O \rangle$ | $\langle s, O \rangle$ | 11.65 | 4.26 | 57.49 | 53.52 | 12.24 | 4.91 | 52.45 | 46.75 |
| 5 | Ours | $\langle s, X, V, O \rangle$ | $\langle s, O \rangle$ | 11.81 | 3.63 | 62.78 | 58.01 | **12.15** | **4.28** | **58.71** | **53.24** |

## A.5 Exhaustive Ablations over Proposed Modules

Our proposed method consists of three modules: (1) object features (2) view aggregation and (3) selective attention. In Table 3, we provide results for all the possible combinations of these three modules. Some combinations of object features, view aggregation, and selective attention are not meaningfully defined. Specifically, view aggregation defines how object features and scene features are combined – without object features there is nothing to combine. We find that our architectural changes (i.e. view aggregation and selective attention) work best in the presence of object features, and that combining all three ideas (row 8) outperforms the other variations.

Table 3: Additional ablations over proposed modules on R2R.

| | Models | Object Features | View Aggregation | Selective Attention | Val Unseen | |
|---|---|---|---|---|---|---|
| | | | | | SR ↑ | SPL ↑ |
| 1 | Baseline | | | | 57.90 | 51.43 |
| 2 | | ✓ | | | 57.26 | 50.96 |
| 3 | | | ✓ | | - | - |
| 4 | | | | ✓ | 57.60 | 52.06 |
| 5 | | ✓ | ✓ | | 57.85 | 51.73 |
| 6 | | ✓ | | ✓ | 57.73 | 52.46 |
| 7 | | | ✓ | ✓ | - | - |
| 8 | Ours | ✓ | ✓ | ✓ | **58.71** | **53.24** |

## A.6 Evaluation Metrics

We describe metrics used in this work below:

– Trajectory Length (TL) measures the average length of agent's trajectory in meters.

– Navigation Error (NE) is the average shortest geodesic distance (in meters) between agent's final location and goal location.

– Success Rate (SR) reports percentage of instructions for which agent's navigation error is less than a threshold. We use threshold of 3 meters for success which is consistent with prior work [5, 7].

– Success weighted by Path Length (SPL) captures the efficiency of agent in reaching goal location. It is obtained by weighing success (1 or 0) by normalized inverse path length factor which is shortest path length divided by maximum of agent's path length or shortest path length.

– Normalized Dynamic Time Warping (NDTW) metric explicitly measures path adherence by softly penalizing deviations from reference path. SDTW provides a "success" analogue to NDTW by multiplying it with success factor (1 or 0) based on distance threshold of 3 meters. We refer the reader to [8] for a detailed discussion on these metrics.

## A.7  VLN↻BERT State Refinement

As discussed in Section 3.3, in VLN↻BERT [6] state history is maintained through the state token $s_t$. Specifically, the next state token $s_{t+1}$ is calculated using the output representation of the state token $\psi(s_t)$, which is refined using the following procedure.

Recall that state-conditioned attention scores over the visual scene features $\alpha(V_t)$ (see Section 3.3) are normalized with the softmax function as $\tilde{a}(V_t) = \mathtt{softmax}(\alpha(V_t))$. These normalized scores are used as next-action probabilities and used to calculate the weighted sum of the scene features $F^v = \tilde{\alpha}(V_t)V_t$.

Similarly, state-conditioned attention scores over the word tokens $\psi(I)$ are calculated as

$$\alpha(\psi(I)) = \frac{\psi(s_t)\psi(I)^T}{\sqrt{d}} \tag{1}$$

and normalized as $\tilde{\alpha}(\psi(I)) = \mathtt{softmax}(\alpha(\psi(I)))$, where $d$ is the model's hidden dimension size. These linguistic scores are used to calculate the weighted sum of word tokens $F^l = \tilde{\alpha}(\psi(I))\psi(V_t)$.

Finally, the next state token is calculated as

$$s_{t+1} = \left[ \left[ s_t; F^v \odot F^l \right] W_1; a_t \right] W_2 \tag{2}$$

where $W_1$ and $W_2$ are learned weight matrices, $\odot$ represents an element-wise product, $[\,\cdot\,;\,\cdot\,]$ represents concatenation, and $a_t$ are directional features for the selected action. Intuitively, this state update procedure aggregates scene, language, and action information into the state history.

## A.8  Qualitative results

In the attached supplemental videos, we visualize a few trajectories from RxR val-unseen split of success and failure cases of our approach along with VLN↻BERT baseline [6]. In each trajectory, we plot panoramic view which is passed as input to the agent and the action which agent takes with an arrow at the bottom. We highlight success with a green box when the agent reaches goal and failure case with a red box.