# OpenReview forum: "SOAT: A Scene- and Object-Aware Transformer for Vision-and-Language Navigation"
_NeurIPS.cc/2021/Conference — NeurIPS 2021 Poster_

### Official Review · Reviewer_7N3z · 2021-07-17

**Rating:** 7
**Confidence:** 4

**Summary:**

This paper proposes a transformer based VLN model based on VLNBERT, which leverages both object-level features (from a Visual Genome pre-trained object detector) and scene-level features (from a Places pre-trained detector). The intuition is that natural language instructions found in the VLN task will contain both references of specific objects in the scene (e.g. chair, sofa), as well as higher level references to entire rooms (e.g. kitchen, bedroom), and that using image features of both types will better the agent’s language grounding. They additionally propose a selective attention mechanism and a view-aggregation strategy. The selective attention only modifies the object-level features through the self-attention layers, keeping the scene-level and instruction features fixed, used for contextualizing the object-level features. The view-aggregation strategy, in my understanding, is a max-pool operation which uses the maximally scoring (compatibility score with the current state) object or scene feature from each view as the view’s representative for action selection. The paper presents a series of experiments comparing the proposed method to prior state-of-the-art in the R2R and RxR datasets, generally beating the presented baselines in the R2R validation-unseen and test splits (with the exception of VLNBERT pre-trained on VLN data) as well as the RxR validation-unseen split. A series of ablations are presented showing that the improved performance is due to the unison of the 3 proposed components rather than any single one alone, and that it rests on vision-language pretraining. Finally, they show that the performance improvement is even larger when evaluating on instances with more than 6 object references.


**Limitations And Societal Impact:**

My only concern with limitations is listed above (about PREVALENT pretraining). I don’t believe the work raises any concerns for potential negative societal impact.


**Main Review:**

Overall: I think the proposed method and motivation behind it are intuitive and will be of interest to the research community. Aside from some minor concerns (detailed below), I think the experiments appear to validate the paper’s claims. I am in favor of acceptance.

Originality: To my understanding, the proposed method is novel and is an extension to already established methods.

Quality: The provided experiments appear to be thorough and clearly show improvements over baselines, with the provided ablations justifying the need for all three proposed components of the method. The only baseline which appears to beat the method is one pre-trained on VLN data, which the model presented does not pre-train on.
My only concerns regarding the experiments are as follows:
* It appears that VLNBERT pre-trained with PREVALENT is the only baseline which beats the proposed method. The Ours condition appears to have been initialized with OSCAR, so I’m wondering if any experiments were run evaluating the performance of the proposed method initialized with PREVALENT? L278-280 comment on this, but I feel like I still wasn’t 100% clear on this. Could the authors please elaborate on this point? More exposition in the paper about why this is orthogonal could make it clearer to readers.
* Were the experiments run with only a single seed? I understand it may not be possible to compare standard deviations to other works which haven’t reported performance across seeds, but it would still be nice to see how stable the presented results are across random seeds.

Clairity: The paper is generally clearly written and easy to follow. Some presentation suggestions:
* L26-27: “The VLN training set” appears to be referencing either R2R or RxR, whereas VLN generally implies the type of task to me. I would suggest referencing the specific dataset here to make this sentence clearer.
* It’s not entirely clear to me what the different sections in the tables mean (e.g. rows 3-7 vs. 8-9 in Table 1), specifying this explicitly in the captions may make the distinction clearer to readers.
* Number bolding appears a little inconsistent. For example, why are some TL numbers bolded and others not? Also got a little confused with multiple numbers being bolded in the same columns (e.g. Table 1), although this seems related to my confusion in the point above about table sections.

* Significance: The proposed method is intuitive and appears to be easy to integrate into existing pipelines -- I believe this will be of value to the research community.


# Post-rebuttal Edit
The response has addressed my concerns, it's also nice to see that performance is consistent across seeds. Following this and reading the other reviews / responses I have decided to keep my score at a 7.

**Time Spent Reviewing:**

5

---

> ### Author Response · Authors · 2021-08-10
> **Author Response to Reviewer 7N3z**
>
> We'd like to thank the reviewer for thoroughly going through our submission and providing detailed feedback, we really appreciate it! We are encouraged to learn that the reviewer found our submission:
>
> - thorough,
> - clear which shows improvements over baselines with the ablations justifying the need for all three proposed components of the method,
> - intuitive and easy to integrate into existing pipelines which will be of value to the research community, and
> - easy to follow and clearly written.
>
> **“The Ours condition appears to have been initialized with OSCAR, so I’m wondering if any experiments were run evaluating the performance of the proposed method initialized with PREVALENT?”**
>
> We did try initializing with PREVALENT weights; however, our initial experiments did not improve with the addition of object features. We hypothesize that this is because PREVALENT was pretrained with scene features on a limited training dataset as compared to the general, object-feature based vision-and-language architectures (e.g., OSCAR) that we build on.
>
> We focused our experiments on the OSCAR pretraining setup because -- similar to other vision-and-language pretraining methods (e.g. LXMERT, UNITER, etc.) -- it was designed for learning generic object grounding from large-scale web data. Our approach uses this grounding to make effective use of object features for VLN. We expect that our findings would extend to other models in this family of pretraining methods.
>
> By contrast, PREVALENT pretraining is not object-based and modifying it to be so is non-trivial and would require innovation beyond the scope of this paper.
>
> **"It would still be nice to see how stable the presented results are across random seeds."**
>
> We agree. We provide results of baseline and our method with multiple random seeds in Table 1 of the supplemental material. The results are consistent with the findings in the main submission.
>
> **Suggested edits.**
>
> We thank the reviewer for the detailed suggestions. We will edit L26-27, update the caption for Table 1, and make the number bolding consistent in the tables.

---

### Official Review · Reviewer_peFD · 2021-07-17

**Rating:** 7
**Confidence:** 4

**Summary:**

This paper proposes an approach for a VLN system based on the Transformer architecture. Building off of existing work, this paper proposes to use both scene and object features of candidate actions (neighboring nodes in the graph). Finding that this doesn't result in improved performance on R2R, the proposed method aggregates the outputs of the Transformer for each candidate action across scene and object representations. The experiments include evaluation on R2R and RxR. There is also analysis of model performance on instructions which include a relatively large number of objects, and it is found that the proposed method has even larger gains on this data.

**Limitations And Societal Impact:**

In comparison to prior work there don't seem to be additional limitations, although the relatively limited experiments make me wonder about the approach's applicability to other environments.

**Main Review:**

Originality: Although the method is new, it seems to be a relatively minor extension of prior work (combining features that prior work only used independently, and a small architecture modification on top of it). The analysis focused on object-heavy instructions is interesting.

Quality: The experiments compare against baselines on two datasets. I would have liked to see experiments on other VLN datasets that also require identifying many objects and scenes, e.g., Touchdown (Chen et al. 2019). Does the proposed approach extend to other environments?

Were experiments ran for multiple trials? I would liked to see significance results on some of the tables. The performance improvements on RxR seem somewhat clear, but it's difficult to tell if the other results are significant. Also, I am not sure why PREVALENT was not experimented with: I would like to see if this method also holds with other pretrained settings, and is not just a result specific to OSCAR (otherwise, it would seem a very specific result with limited applicability). Ablating scene and object features independently would be interesting to see.

Clarity: The paper was easy to read and the method was clearly explained.

Significance: Because this method was evaluated only on one environment (two datasets), and seems to rely heavily on pretraining setup, it is a bit unclear to me how extensible it is to other VLN datasets. I am not sure what the general takeaway from this paper is besides the fact that using multiple sources of input which prior work did not combine before improves model performance if the architecture is slightly modified.

**Time Spent Reviewing:**

1

---

> ### Author Response · Authors · 2021-08-10
> **Author Response to Reviewer peFD**
>
> We'd like to thank the reviewer for the insightful feedback and acknowledging that our submission was easy to read and the method was clearly explained. We take this opportunity to provide clarifications to the concerns raised below.
>
>
> **“Does the proposed approach extend to other environments?”** and **“e.g., Touchdown (Chen et al. 2019)”** and **“Because this method was evaluated only on one environment (two datasets), and seems to rely heavily on pretraining setup, it is a bit unclear to me how extensible it is to other VLN datasets. I am not sure what the general takeaway from this paper is.”**
>
> The general takeaway is that VLN performance can be improved by encoding the inductive bias that the world is composed of objects. We believe this should generalize to other environments (like Touchdown) because they too contain instructions referring to objects.
>
> We note that the components of our model (i.e., the visual encoders and the multimodal transformer) were pretrained on a diverse collection of web data (e.g., Places365, VisualGenome, Conceptual Captions) that is not specific to the indoor navigation datasets that we experimented with in this work (i.e., R2R and RxR). This pretraining data includes both indoor and outdoor imagery. Thus, just as this pretraining is useful for the indoor setting (see section 5.2 in our main paper), it should generalize to other settings as well.
>
> We agree that exploring performance on other datasets is important. However, we believe it is beyond the scope of this paper. We will open-source our code for easy adoption to other vision-and-language navigation datasets by the community.
>
> **“Also, I am not sure why PREVALENT was not experimented with: I would like to see if this method also holds with other pretrained settings, and is not just a result specific to OSCAR”**
>
> As discussed in Sections 2 (L93-95), 1 (L61-63), and 5 (L299-L301), we experimented with OSCAR because -- similar to other vision-and-language pretraining methods (e.g. LXMERT, UNITER, etc.) -- it was designed for learning generic object grounding from large-scale web data. Our approach uses this grounding to make effective use of object features for VLN. We expect that our findings would extend to other models in this family of pretraining methods.
>
> In contrast, PREVALENT pretraining is not object-based and modifying it to be so is non-trivial and would require innovation beyond the scope of this paper. We did try initializing with PREVALENT weights; however, our initial experiments did not improve with the addition of object features. We hypothesize that this is because PREVALENT was pretrained with scene features on a limited training dataset as compared to the general, object-feature based vision-and-language architectures (e.g., OSCAR) that we build on.
>
> **“Were experiments run for multiple trials?”**
>
> Yes. We provide results for multiple seeds in Table 1 in the supplemental material. The results are consistent with those reported in the paper.
>
> **“Ablating scene and object features independently would be interesting to see.”**
>
> We agree. We provided an ablation using only object features in the supplemental material in Table 2 row 4, which demonstrates the importance of using both scene and object features in our proposed approach.

---

### Official Review · Reviewer_2Xzi · 2021-07-18

**Rating:** 7
**Confidence:** 3

**Summary:**

This paper presents a method for the vision-and-language navigation task, where an agent needs to navigate through a set of rooms to a goal location as specified by an instruction. What this paper focuses on is making use of object-features in addition to scene features in the visual input. Having both explicit in the architecture covers objects and scenes that are often used in the instructions and allows the method to leverage pre-training of these representations. The paper finds that simply adding the object features to the input alone does not improve performance, which instead requires additional changes to the architecture, i.e. selective attention and view aggregation. With these changes in place, the paper shows improved performance, especially for instructions that heavily refer to objects.

**Limitations And Societal Impact:**

No issues here.

**Main Review:**

# Originality

The paper builds on existing approaches to the vision-and-language navigation task based on multimodal transformers and extends them to take objects into account. It is clear how this paper differs from prior work and credit to earlier work is adequately given. The paper does an excellent job in clearly separating novel contributions from prior work. Well done!

The paper is somewhat incremental but I do not see this as a negative point. The proposed approach is an instantiation of a logical next step and provides valuable insights that others can build on in turn.

# Quality

The submission seems technically sound. All claims are well supported and the used methods and baselines are appropriate. The work seems complete and the comparison to SOTA seems fair.

The main weakness I see in this work is that I do not feel confident in having understood the relative importance or the interaction of the three main contributions (object features, view aggregation, and selective attention). The ablation analysis (Table 3) progressively adds the three contributions to the baseline and evaluates these versions, but based on these results we can really only be confident that the last of the additions contributes to improved performance, as the other versions (lines 2-3) are not significantly better than the baseline. To get a better understanding of the influence of the other components, we would need all 8 combinations of using or not using a component (only 2x the experiments that are already done) – and if that is not possible, removing the components one at a time would be preferable over the chosen setup. In addition to verifying that each contribution is valuable, it would be important to show that they interact and that the view aggregation and the selective attention enable the use of objects as claimed by the authors. With the current results, one explanation for all the datapoints would be that the object input generally hurts performance (except in the heavy object case) but that the other contributions independently lead to performance improvements so that the overall performance is better than the baseline.

The differences between the seen and unseen environments in Table 3 also point to the fact that what is happening here might mainly be a question of proper regularization. If that is the case it might be less important which modes (instruction, scene, objects) are selected in the selective attention but rather the fact that attention is selective in some way. To verify whether this is the case, it would be good to try out other versions of selective attention as well as other approaches that would regularize the model.

# Clarity

The submission is very clearly written and well organized. The background section (Section 3) provides enough context to make this work very accessible. With the clear description and the provided code, it should be straightforward to reproduce the work. Well done!

# Significance

The paper presents a significant step in the vision-and-language navigation task combining the use of both the scene and objects. If all proposed components are properly analyzed (see above), this paper should enable follow-up work on this topic. Unfortunately, the current evaluation does not fully convince me that all proposed components are actually necessary or that they interact in a way that the performance improvement is mainly driven by object-awareness as claimed by the paper. This experimental analysis needs to be improved before I can recommend accepting this paper.

# Post-Rebuttal EDIT

My main concerns have been addressed by improving the analysis of the proposed components.


**Time Spent Reviewing:**

2

---

> ### Author Response · Authors · 2021-08-10
> **Author Response to Reviewer 2Xzi**
>
> First of all, we'd like to thank Reviewer 2Xzi for mentioning that our submission:
> - is technically sound,
> - seems complete and the used baseline and approach are appropriate,
> - is an instantiation of a logical next step and provides valuable insights that the research community can build on,
> - is very clearly written and well organized, and the background section provides enough context to make this work very accessible.
>
> We take this opportunity to clarify some of the concerns raised.
>
> **Understanding the relative importance of object features, view aggregation, and selective attention -- need experiments for all 8 combinations.**
>
> We appreciate the reviewer's concerns and have run the exhaustive set of ablations (showing all 2^3 = 8 possible combinations of the 3 ideas). We find that our architectural changes work best in the presence of object features, and that combining all three ideas (row 8) outperforms the other variations by 0.78 - 2.28 SPL. However, we want to clarify that some combinations of object features, view aggregation, and selective attention are not meaningfully defined. Specifically, view aggregation defines how object features and scene features are combined -- without object features there is nothing to combine. We present the ablations below including those from the original table.
>
> | # | object features | view aggregation | selective attention | val unseen SR | val unseen SPL |
> | --- | :---: | :---: | :---: | --- | --- |
> | 1 (baseline) |  |  |  | 57.90 | 51.43 |
> | 2 | ✔ |  |  | 57.26 | 50.96 |
> | 3 (undefined) |  | ✔ |  | undefined | undefined |
> | 4 |  |  | ✔ | 57.60 | 52.06 |
> | 5 | ✔ | ✔ |  | 57.85 | 51.73 |
> | 6 | ✔ |  | ✔ | 57.73 | 52.46 |
> | 7 (undefined) |  | ✔ | ✔ | undefined | undefined |
> | **8 (ours)** | ✔ | ✔ | ✔ | **58.71** | **53.24** |
> |  |  |  |  |  |  |
>
> **Try other attention patterns.**
>
> We were also curious if other attention patterns would improve performance, so we tried a number of variations and provided results in Table 2 of the supplemental material. None of the alternative attention patterns outperformed our proposed approach.
>
> Moreover, the ablation results above demonstrate that selective attention alone (row 4) does not lead to the best performance. Instead, combining selective attention with object features and view aggregation (row 8) results in the largest gains.

---

> > ### Comment · Reviewer_2Xzi · 2021-08-16
> > **Re: Response**
> >
> > I am very happy to see my main concerns addressed and the paper improve as a result. I will update my rating accordingly.

---

### Decision · Program_Chairs · 2021-09-27

**Decision:**

Accept (Poster)

**Comment:**

This paper presents a method for vision and language navigation based on transformers and leveraging information from object-level processing.

The paper received favorable reviews and a consensus emerged very quickly on acceptance, in particular taking into account a convincing rebuttal.

The AC concurs.